# An Engineering Method for Resonant Microcantilever Using Double-Channel Excitation and Signal Acquisition Based on LabVIEW

**DOI:** 10.3390/mi14040823

**Published:** 2023-04-07

**Authors:** Shanlai Wang, Zhi Cao, Xiaoyang Zhang, Haitao Yu, Lei Yao

**Affiliations:** 1School of Microelectronics, Shanghai University, Shanghai 200444, China; wangshanlai@shu.edu.cn (S.W.); kanolin@shu.edu.cn (X.Z.); 2State Key Laboratory of Transducer Technology, Shanghai Institute of Microsystem and Information Technology, Chinese Academy of Sciences, Shanghai 200050, China; 216062101@mail.sit.edu.cn (Z.C.); yht@mail.sim.ac.cn (H.Y.); 3School of Chemical and Environmental Engineering, Shanghai Institute of Technology, Shanghai 201418, China

**Keywords:** resonant microcantilever, TGA, dual-channel, LabVIEW

## Abstract

Resonant microcantilevers have the advantages of ultra-high heating rates, analysis speed, ultra-low power consumption, temperature programming, and trace sample analysis when applied in TGA. However, the current single-channel testing system for resonant microcantilevers can only detect one sample at a time, and need two program heating tests to obtain the thermogravimetric curve of a sample. In many cases, it is desirable to obtain the thermogravimetric curve of a sample with a single-program heating test and to simultaneously detect multiple microcantilevers for testing multiple samples. To address this issue, this paper proposes a dual-channel testing method, where a microcantilever is used as a control group and another microcantilever is used as an experimental group, to obtain the thermal weight curve of the sample in a single program temperature ramp test. With the help of the LabVIEW’s convenient parallel running method, the functionality of simultaneously detecting two microcantilevers is achieved. Experimental validation showed that this dual-channel testing system can obtain the thermogravimetric curve of a sample with a single program heating test and detect two types of samples simultaneously.

## 1. Introduction

Thermogravimetric analysis (TGA) [1,2] is a classic characterization method widely used in various fields such as new material development [3,4,5,6,7], chemistry [8,9], physics [10,11,12], and energy [13,14,15]. The TGA method enables measurement of the material’s structure [16,17], composition [8,18], thermal stability [3,19,20], and reaction kinetics [2,21,22], among other information. The core of existing TGA instruments is a precision microbalance, which has a mass detection sensitivity of sub-microgram levels and can thus record the mass loss of milligram-level samples in real-time. However, with the rapid development of new functional materials and related research fields, the technical limitations of existing TGA methods and instruments are becoming increasingly prominent, and they are unable to meet the requirements for accurate measurement [23,24,25]. It is worth noting that the modern thermal balance instrument has a quality sensitivity at the sub-microgram level, while the sample consumption is at the milligram level. Therefore, existing TGA instruments require a large amount of sample, leading to significant waste of sample materials. In addition, it is difficult to uniformly heat large volume samples during rapid heating, making it difficult to obtain accurate results. Moreover, the analysis time is also very time consuming, resulting in low efficiency of TGA experiments, which cannot meet the high-efficiency development needs of new materials [26,27]. This is a technical challenge that cannot be solved by existing TGA instruments. In addition, the large quantity of samples makes it too risky to analyze hazardous samples or substances containing energy (such as strong oxidizers and explosives). Existing TGA instruments are prone to damage from exothermic chemical reactions (such as explosions, strong oxidation, or corrosion) induced by heat during the TGA process. Furthermore, commercial TGA instruments use bulky enclosed furnaces to heat samples with relatively large volumes, making it difficult to achieve in situ observation of material phase or structural evolution. It is also challenging to combine them with other analytical techniques, such as Raman spectroscopy [28,29], to meet the growing demand for real-time characterization [30]. Therefore, the application scope and functional versatility of existing TGA instruments are severely limited.

Yao et al. [31] proposed an integrated silicon resonant microcantilever as an ultra-sensitive micro-thermal analysis chip for next-generation thermogravimetric analysis (TGA) characterization. This chip is called a micro-electromechanical system based TGA (MEMS-based TGA or MEMS TGA). Compared to commercially available TGA instruments, the MEMS TGA has a series of significant advantages, such as requiring a small demand for trace samples, achieving ultra-high resolution mass measurement, ultra-high heating rate and analysis speed, and ultra-low power consumption. The principle of the MEMS TGA is to reflect the real-time mass change (Δ*m*) by the resonance frequency change (Δ*f*) of the resonant microcantilever [32,33]. Currently, the testing process of this MEMS TGA is to perform two program heating cycles (one called the baseline and the other called the weight loss curve) for each sample and obtain the corresponding TGA curve through the corresponding algorithm. However, as only one microcantilever can be used for each test, it is time consuming to perform consecutive two-program heating cycles, and the efficiency is low since only one sample can be tested at a time.

To address the issue of single-channel testing in the current MEMS TGA, this paper proposes a multi-channel testing technology that can simultaneously use multiple microcantilevers for testing. To achieve the detection of the resonance frequency signals of multiple channels of microcantilevers, an interface circuit system is designed, and the core of the system is a phase-locked loop (PLL) [34]. Due to the limitations of the analog circuit phase-locked loop, this paper adopts a LabVIEW-based software phase-locked loop [35,36,37,38], which can quickly and accurately track the changes in the resonance frequency and avoid the complexity of the analog phase-locked loop.

To verify the feasibility of the multi-channel resonant microcantilever testing system, and taking advantage of the parallel operation mode of LabVIEW which is conducive to implementing multi-channel data acquisition [39,40,41,42], this paper developed a LabVIEW-based dual-channel detection system for resonant microcantilevers. With only one measurement, the dual-channel detection system can complete TGA testing (one channel as baseline, and the other channel as detection). Furthermore, as two channels can simultaneously test two different samples, this technology can greatly improve the testing efficiency of MEMS TGA and be extended to more channels, demonstrating great potential for high-throughput detection applications.

## 2. System Structure and Principle

The whole system is composed of a resonant microcantilever, a hardware circuit, and software code. The three components’ functioning is explained in the following.

Figure 1 displays the structure of the entire system. LabVIEW on a PC can set parameters in accordance with the needs of the experiment as the system’s control hub. The NI data acquisition card then generates control words in accordance with the instructions and sends them to the Direct Digital Frequency Synthesizer after receiving parameter instructions from LabVIEW. To drive the resonant microcantilever vibration, the Direct Digital Frequency Synthesizer generates the appropriate excitation signal in accordance with the control word. Excitation and reaction signals are gathered by the NI data acquisition card and sent to LabVIEW on the PC for computation. The frequency of excitation pulses is then modified in accordance with the information collected.

### 2.1. The Operating Principle of a Resonant Microcantilever

The core of the MEMS TGA is the resonant microcantilever. The specific structure of the microcantilever is shown in Figure 2. The resonant microcantilever mainly consists of a fixed silicon base and a cantilevered plate-like structure, which can be regarded as a resonator. This paper adopts the method of using electric heating to make the resonant microcantilever work normally, and uses a circuit with piezo-resistive detection to collect the vibration (pick-up) of the resonant microcantilever. The front end of the cantilever is the sample area and the heating resistor. Before the heating resistor and the piezo-resistive resistor, there is a specially designed square window, which can effectively prevent the heat from the heating end from being conducted to the fixed end.

The resonant microcantilever utilizes the principle of mechanical energy conservation during resonance to output a resonance frequency shift (Δ*f*) signal, representing real-time mass changes (Δ*m*). An electric resistance-based microheater is designed at the free end of the vibrating beam for sample heating, achieving heating temperatures of up to 1200 °C. By using pre-calibrated metal heating resistors as temperature sensors to control heating current, programmable temperatures can be achieved. The mass loss during the programmed heating process is measured using a piezo-resistive Wheatstone bridge to read the frequency shift signal in real time. The piezo-resistive sensor element and the thermoelastic resonant excitation element are both integrated on the microcantilever, with resonance maintained through a phase-locked loop interface circuit. The weight loss detection resolution of the resonant microcantilever can reach 1 pg (10^−12^ g), with only nanogram-level (10^−9^ g) samples required for conventional MEMS TGA measurements. The nanogram-level samples can be uniformly heated by the microheater located at the end of the microcantilever. Compared to the non-uniform temperature distribution typically required for milligram-level samples in traditional TGA, the trace samples in MEMS TGA can always be uniformly heated, achieving lower thermal lag even at very high heating rates.

### 2.2. Design of Hardware Structures

The drive circuit and signal conditioning circuit are the major components of the system’s hardware.

#### 2.2.1. Drive Circuit

The Direct Digital Frequency Synthesizer and variable gain amplifier modules can be separated into the system’s drive circuit.

In 1971, American researchers J. Tierney et al. proposed the Direct Digital Frequency Synthesizer (DDS) [43], a novel digital frequency synthesizer device. DDS is appropriate for the system proposed in this study because it can carry out operations using rapid frequency conversion, phase digital regulation, high resolution, cheap cost, and low power consumption. To excite the resonant microcantilever, the PC-based LabVIEW program synthesizes the AC signal of the required frequency and feeds it via the data acquisition card to the DDS module. The AD9850 DDS chip is used in this paper. In Figure 3, the circuit diagram is displayed. A 0.5 V sinusoidal signal is output through the Z_OUT pin in Figure 3 using serial communication (pin W_CLK, FQ_UD, D7, and RESET).

#### 2.2.2. Circuit for Signal Processing

The signal conditioning circuit includes a differential amplifier circuit and band-pass filter module because the signal collected by the Wheatstone bridge on the resonant microcantilever is very weak and prone to noise and interference. This means that in order to extract the useful output signal, the signal must be amplified and filtered.

In this paper, the precision instrument amplifier AD8422 is used. It is a type of instrument amplifier that has excellent accuracy, minimal power usage, and minimal noise from rail to rail. The AD8422 can vary gain from 1 to 100 with just one gain resistor, and has a maximum static current of 330 μA and a maximum input voltage noise of 8 nV/Hz at 1 kHz, making it perfect for conditioning piezo-resistive Wheatstone bridges. Figure 4 depicts the operational amplifier circuit for the AD8422. The gain of the AD8422 can be adjusted by crossing a gain resistor to the *R_G_* pin. The following is the gain calculation formula:(1)$$R_{G}=\frac{19.8\;k\Omega }{G-1}$$

The gain circuit can be calculated using the Formula (1) as *R_G_* = 2.2 kΩ, when *G* = 10.

After the AD8422 amplifies the resonant microcantilever’s output signal, a band-pass filter filters out the noise brought on by resistance imbalance and temperature drift in bridge. The operational amplifier OP37, which is low-noise, precise, and fast, is employed in this paper to filter the noise. Figure 5 depicts the schematic diagram of this circuit.

### 2.3. Design of Software Structures

The computer serves as the system’s control center and is responsible for storing data and transmitting commands to govern each module. This paper’s system is a graphical control software created using LabVIEW. The user interface, information presentation, parameter settings, and data storage and reading are among its primary features. The ability to generate control words, open loop tests, closed loop tests, and data preservation make up the entirety of the software.

#### 2.3.1. Generate Control Words Module

The DDS typically includes a phase accumulator, ROM, frequency reference source (often a crystal oscillator), and a DAC. The phase accumulator accumulates the input frequency control word (FCW) throughout each clock cycle to determine the proper phase angle of the output sine wave [44]. The DDS’s output frequency is
(2)$$f_{out}=f_{clk}\times \frac{FCW}{2^{N}}$$
where *N* is the size of the phase accumulator, *FCW* is the input frequency control word, and *f_clk_* is the clock frequency of the DDS.

To generate the appropriate frequency control word to deliver to the DDS, LabVIEW must send instructions to the NI data acquisition device. The control word generation portion of the program displayed in Figure 6a is written based on the aforementioned guidelines. Figure 6b shows that child VI of a is b. In this work, the system is chosen to have a 125 MHz crystal oscillator, and N is chosen to have 32 bits.

The system requires four digital pins for each channel. The pin out of the NI USB-6361 is shown in Figure 7, and eight digital pins from P0.0 to P0.7 have been chosen. The first channel’s RESET, FQ_UD, CLK, and DATA ports are represented by pins P0.0 to P0.3. The corresponding ports of the second channel are P0.4 through P0.7.

#### 2.3.2. Open Loop Test

The objective of the open loop test is to determine the amplitude–frequency curve and phase–frequency curve of the resonant microcantilever sensor that will be put to the test using the method of sweeping frequency. This will allow for the accurate determination of the two key parameters of resonant frequency and resonant phase difference for the closed loop test. The procedure is described below.

Figure 8 describes the open loop test flow. After setting the test parameters, the NI data acquisition card receives instructions from LabVIEW and generates control words, which are then transmitted to the DDS. Based on the time interval Δ*f* and the received control word, the DDS generates an incentive signal from *f_start_* to *f_end_*. The computer determines the value of Δ*f* and the precision threshold by starting with the frequency that corresponds to the highest value of the current frequency interval, or *f*_0_. If Δ*f* is less than or equal to the precision threshold, *f*_0_ is the resonant frequency and LabVIEW records *f*_0_, *ϕ*_0_ and the frequency–phase difference coefficient *k*, and *ϕ*_0_ is ready to enter the closed loop test. The parameter is altered if Δ*f* is bigger than the precision threshold, and the formula is as follows:(3)$$f_{start}=f_{0}-\frac{f_{end}-f_{start}}{20}\;f_{end}=f_{0}+\frac{f_{end}-f_{start}}{20}\;\Delta f=\frac{\Delta f}{10}$$

Repeat the previous steps until is below or equal to the precision level. The program diagram’s open loop test section is displayed below.

The program in Figure 9a performs the function of extracting the frequency value *f*_0_ (resonant frequency) corresponding to the highest value in a frequency interval of each channel and the accompanying phase *p*_0_. The variables “Harmonic Analysis-C1” and “Harmonic Analysis-C2” represent the harmonic information for each channel, and their data including frequency, amplitude, and phase difference can be extracted using the “Index Array” control. The amplitude maximum and its corresponding index position can be found using the “Array MAX&MIN” control, and the frequency and phase difference corresponding to this index position can be matched and obtained. The program’s purpose in Figure 9b is to establish Δ*f* and the precision threshold, as well as modify the sweep test parameters. Initially, a “Case Structure” is employed to determine whether Δ*f* is less than or equal to the accuracy threshold. If not, a “Stacked Sequence Structure” is used, which is responsible for the implementation of the variable calculation based on Equation (3) as depicted in the figure.

Figure 10 displays the user interface (UI) for harmonic analysis of two channels.

#### 2.3.3. Closed Loop Test

The resonant microcantilever and the software phase-locked loop are used in the closed loop test to track the resonant frequency in real time.

Software phase-locked loops are at the heart of closed loop testing. Excitation signal modification and phase difference extraction using the DFT algorithm make up the software phase-locked loop. The goal is to maintain a consistent phase difference between the two signals by synchronizing the DDS excitation signal’s frequency and phase with the resonant microcantilever’s output signal [45].

The diagram below depicts the closed loop test’s flow. The first excitation frequency used to generate the resonant microcantilever vibration is the resonant frequency *f*_0_ discovered during the open loop test. Then use the DFT method with the gathered response and excitation signals to determine the phase different *ϕ*. Next, determine the difference between the open loop test’s phase difference *ϕ*_0_ and resonant phase difference *ϕ*. Then the new resonant frequency is computed using the phase difference–frequency relation coefficient *k* discovered during the open loop test. This realizes the tracking frequency function in real time.

The primary flow of closed loop testing is depicted in Figure 11a. The software phase-locked loop computation excitation signal program block diagram is in (b); the phase difference between the excitation signal and the response signal is calculated using a software phase-locked loop block diagram (c). The “Array” variable retrieves information about the resonant frequency of the simulated signal during closed-loop testing, and then generates the corresponding waveform using “Build Waveform”. The waveform information is then written to the “Compute Phase Difference” sub-VI in (c), where the phase difference is calculated using the DFT algorithm. The resulting phase difference is then used to calculate the frequency shift by subtracting the resonant phase difference obtained from open-loop testing, denoted by *ϕ*_0_. This calculation is performed using the phase difference–frequency coefficient *k* obtained from open-loop testing, which yields the resonant frequency.

Figure 12 shows the UI for closed loop testing, which primarily comprises frequency–time plots of two channels and real-time display of various data.

#### 2.3.4. Data Preservation

The data preservation module is mainly responsible for the real-time storage of the frequencies collected from two channels during testing. Its main program architecture is shown in Figure 13.

The sequential numbering of the frequencies collected from two channels is achieved through the use of a shift with a “while loop” and a “+1” control. The “while loop” contains a flat sequential structure, in which the first frame writes the frequencies of the two channels to the corresponding paths in a text file, while the second frame provides a delay function that determines the time delay based on the user-defined variable “save count”.

## 3. Results and Discussion

The dual-channel test system for resonant microcantilever is designed in this study using LabVIEW in accordance with the aforementioned guidelines, and two experiments are designed: (1) To confirm that the system can perform difference control, in channel one, a resonant microcantilever without any sample (bare beam) is mounted; in channel two, a resonant microcantilever with CaC_2_O_4_·H_2_O (sample beam) is used. They are heated simultaneously. Finally, the weight loss rate is determined by integrating the information from the sample beam and bare beam. (2) To evaluate the accuracy of the system, a resonant microcantilever containing CaC_2_O_4_·H_2_O samples was used for each of the two channels and then heated at the same time. The samples’ rates of weight loss are then calculated, and the results from the two sets of data are contrasted.

### 3.1. Experiment 1

This test is designed to determine whether differential comparison can be utilized to derive the thermogravimetric curves of two resonant cantilevers. The following actions are specific to the test performed.

(1)To burn out contaminants on the empty beam, the two channels’ empty beams are heated simultaneously from ambient temperature to 800 °C for 5 min.(2)Measure the frequency of the beams under empty conditions, and calculate the channel with a hollow beam using two empty beam frequencies’ difference, remembering to consider Δ*F*.(3)Run a temperature ramp program on the empty beam of channel two with the following parameters: starting temperature of 50 °C, ending temperature of 800 °C, heating rate of 20 °C/min. After the temperature ramp program is completed, save the frequency file as a validation baseline for later use.(4)Add the CaC_2_O_4_·H_2_O sample to the empty beam of channel two and bake in an 80 °C oven for 10 min.(5)Take out the baked sample beam from the oven and insert it into the device, then simultaneously perform a heating program with the following parameters: starting temperature of 50 °C, ending temperature of 800 °C, heating rate of 20 °C/min. After the heating program is completed, save each frequency file separately.(6)Use the frequency file of the empty beam in channel one as the baseline; then, add Δ*F* to each moment’s frequency *F* in the frequency file to the sample beam in channel two, save it as a new frequency file, and use it as the sample weight loss frequency line.(7)Combine the baseline from channel one with the sample weight loss frequency line calculated from channel two to compute the sample thermal gravimetric curve, and compare it against the standard.

Experiment 1 was primarily conducted to validate the accuracy of the dual-channel testing method proposed in this study. Figure 14a shows the thermogravimetric curves obtained from the single-channel and dual-channel testing methods, revealing that both curves exhibit approximately 12.9%, 20.6%, and 27.2% weight loss in the first, second, and third stages, respectively, resulting in a total weight loss of approximately 60.7%. Figure 14b illustrates the error between the thermogravimetric curves obtained using the dual-channel testing method and those obtained using the single-channel testing method, with a difference of less than 0.1%. The experimental results demonstrate that the system designed in this study is capable of differentially controlling the reference and test channels and accurately measuring the results obtained from both channels.

### 3.2. Experiment 2

The following steps are performed for testing after the experimental tools and samples are prepared:(1)To burn out contaminants on the empty beam, the two channels’ empty beams are heated simultaneously from ambient temperature to 800 °C for 5 min.(2)Two empty beams simultaneously experience their predetermined temperature rise. The following are the details of the temperature rise: starting temperature of 50 °C, ending temperature of 800 °C, heating rate of 20 °C/min. The frequency files are saved as their baselines after the application has been heated.(3)Samples of CaC_2_O_4_·H_2_O should be added to the two empty beams before being baked for 10 min at 80 °C.(4)Two sample beams are removed and set up to increase in temperature. The following are the parameters for the temperature rise: starting temperature of 50 °C, ending temperature of 800 °C, heating rate of 20 °C /min. The frequency files are preserved as their weightless frequency lines once the application has heated them up.(5)The two resonant microcantilevers’ TG curves are calculated and evaluated separately.

Figure 15 shows the thermalgravimetric curves of two channels. As can be seen from the curves, both channels lost about 12.5% in the first stage; 20.9% in the second stage; 27.9% in the third stage; and about 61.3% in total. The experimental results demonstrate that the dual-channel system designed in this study is capable of simultaneously testing two samples.

## 4. Conclusions

This article mainly discusses the principles, implementation process, and experimental verification of the double-channel testing system for the resonant microcantilever.

This study proposes a dual-channel testing method in response to the limitations of the current resonant microcantilever system’s single-channel testing. The method mainly employs differential control, with one channel acting as the control group and the other as the experimental group. From experiment 1, it can be observed that the proposed dual-channel testing method is feasible and provides a 50% increase in efficiency compared to the single-channel testing method. As the testing process can be completed with only one program heating cycle, the longer the required program heating time, the more time can be saved by this method. At minimum, a 50% reduction in time can be achieved. Moreover, the microcantilever in the control group can be reused, reducing device consumption. It is worth nothing that the dual-channel testing method can synchronize the baseline and sample weight loss curves in identical environments, providing more rigorous experimental data. Experiment 2 demonstrates that the designed dual-channel system can simultaneously test two microcantilevers with samples, greatly improving testing efficiency. In fact, in some respects, compared to the single-channel method, the dual-channel testing method can obtain test results for two samples in the same amount of time, resulting in an efficiency increase of approximately 100%, and laying the foundation for the implementation of more channels, with the potential for high-throughput detection applications.

## Figures and Tables

**Figure 1 micromachines-14-00823-f001:**
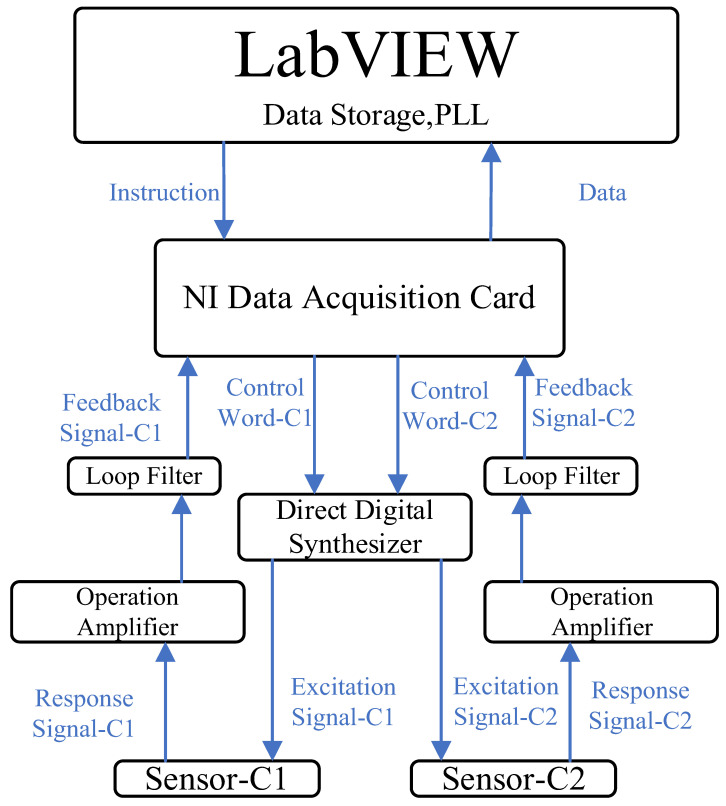
System architecture diagram.

**Figure 2 micromachines-14-00823-f002:**
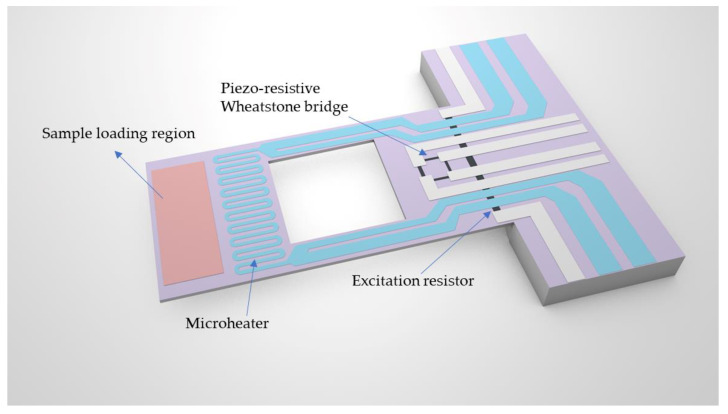
Diagram of resonant microcantilever structure.

**Figure 3 micromachines-14-00823-f003:**
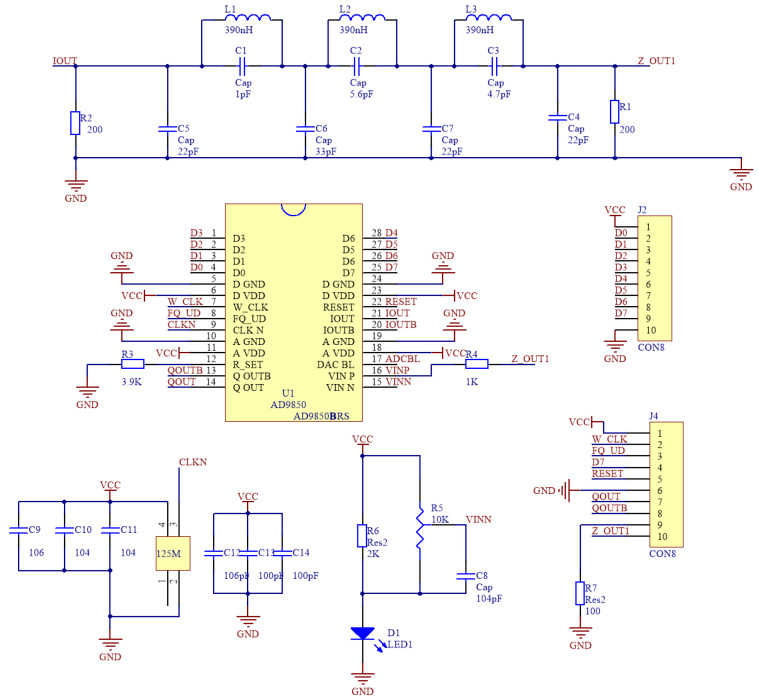
Circuit diagram of AD9850-based DDS.

**Figure 4 micromachines-14-00823-f004:**
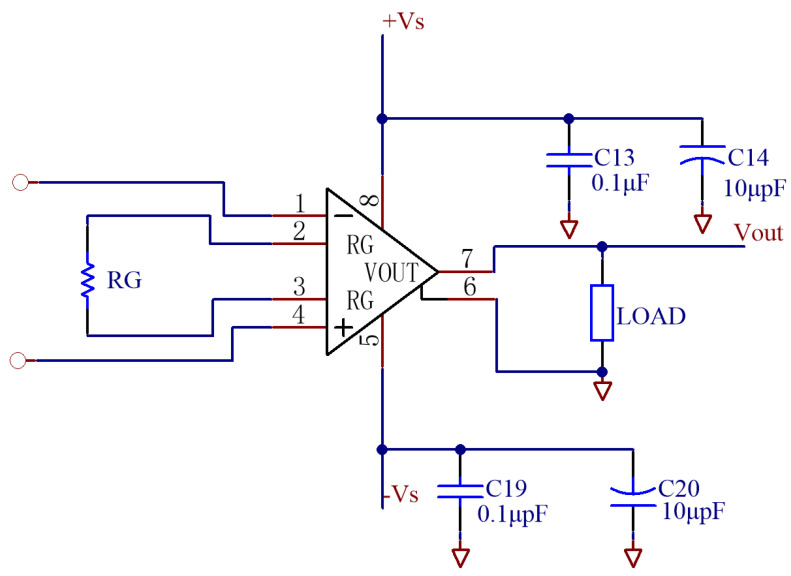
AD8422-based operational amplifier circuit.

**Figure 5 micromachines-14-00823-f005:**
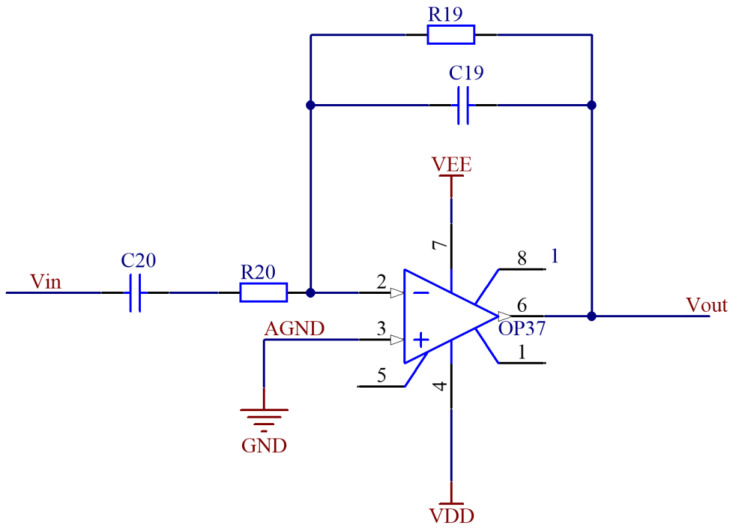
OP37-based band-pass filter.

**Figure 6 micromachines-14-00823-f006:**
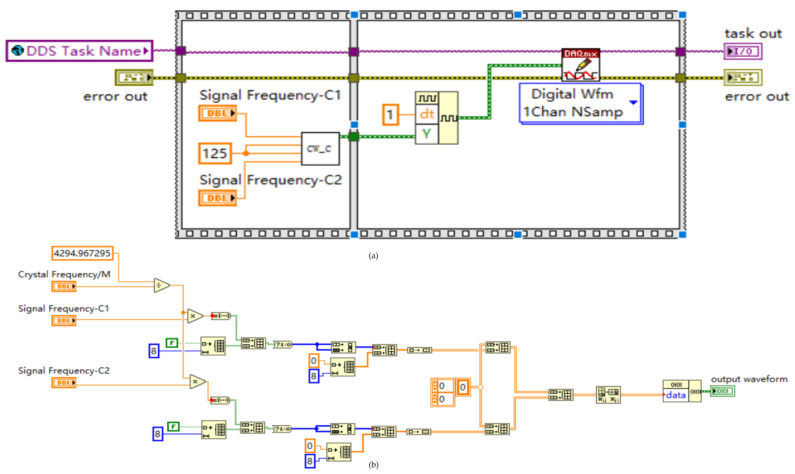
Writing a control program. (**a**) Describes a program that completes the task of generating the DDS control word. (**b**) Describes a sub-VI that calculates the control word based on Equation (2).

**Figure 7 micromachines-14-00823-f007:**
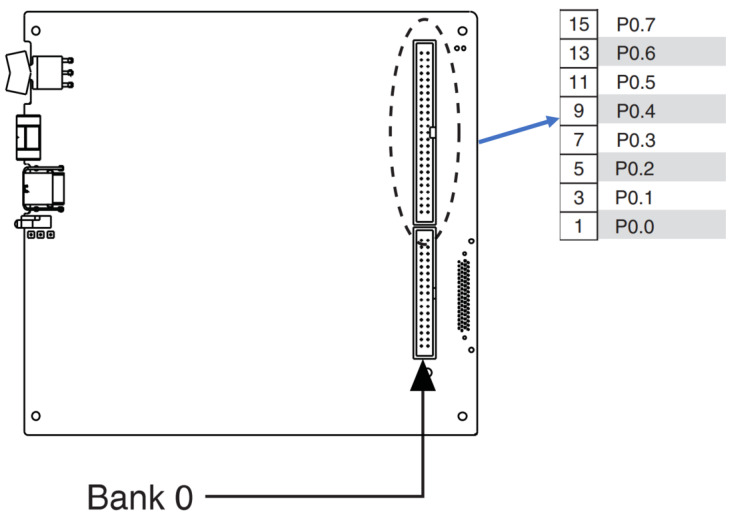
Some of the pins of the NI-6361 data acquisition card.

**Figure 8 micromachines-14-00823-f008:**
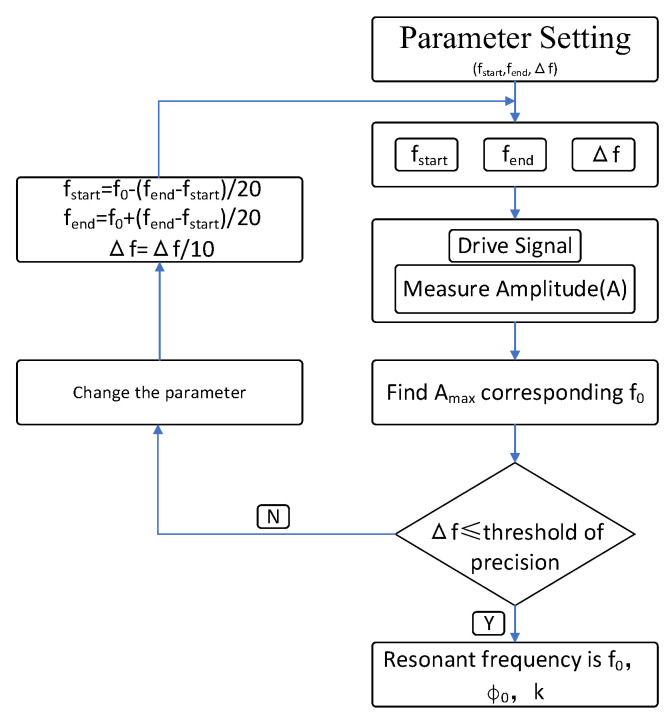
Open loop test flow chart.

**Figure 9 micromachines-14-00823-f009:**
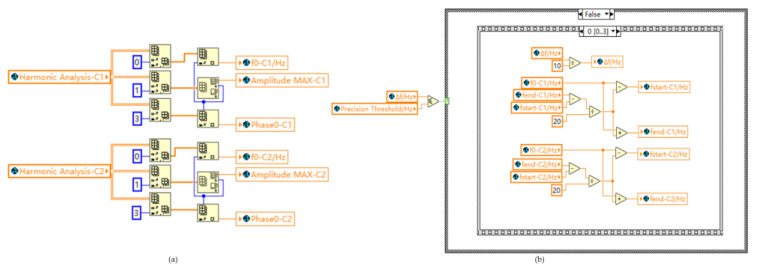
Open loop test program flow chart. (**a**) Program flow chart for obtaining *f*_0_ in the open loop test. (**b**) Flow chart for adjusting parameters in the open loop test.

**Figure 10 micromachines-14-00823-f010:**
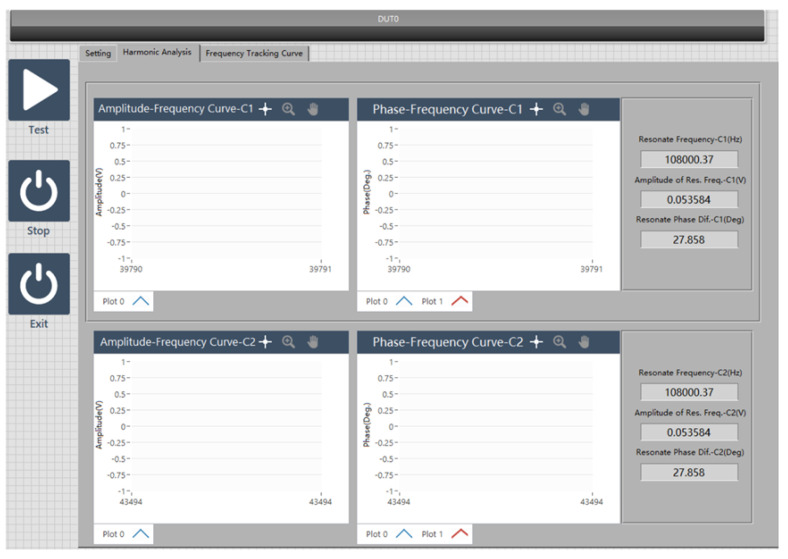
UI for harmonic analysis.

**Figure 11 micromachines-14-00823-f011:**
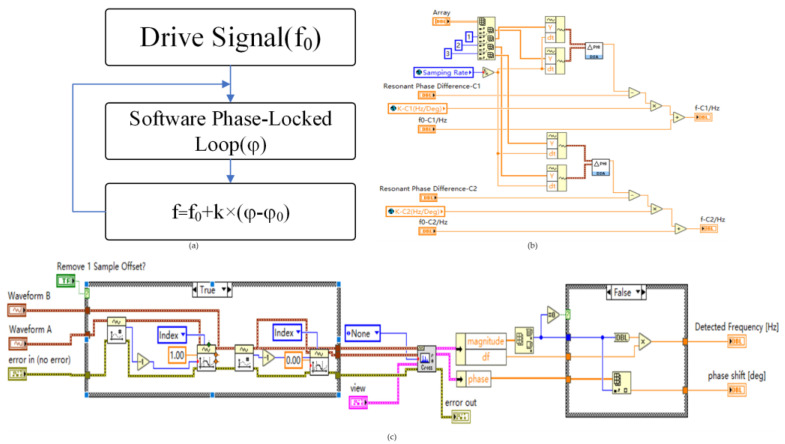
Closed loop test flow chart and partial program flow chart. (**a**) Flow of the closed loop test. (**b**) Program for calculating the excitation frequency through software phase-locking. (**c**) Sub-VI for extracting the phase difference between the excitation signal and the response signal in the software phase-locking loop.

**Figure 12 micromachines-14-00823-f012:**
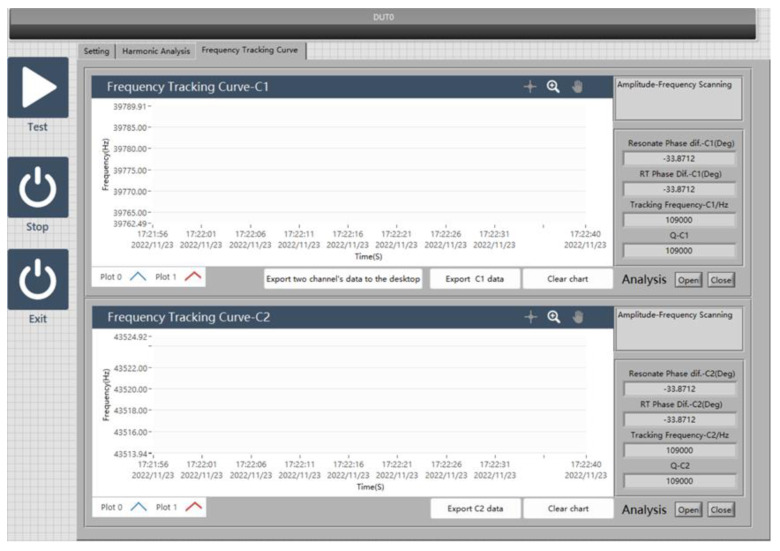
Closed loop testing of the user interface.

**Figure 13 micromachines-14-00823-f013:**
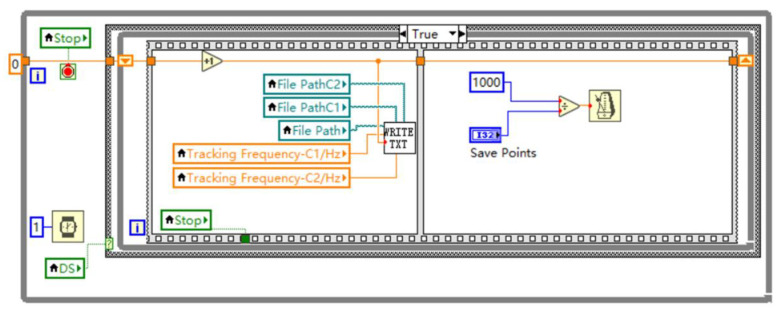
Partial program code of the data preservation module.

**Figure 14 micromachines-14-00823-f014:**
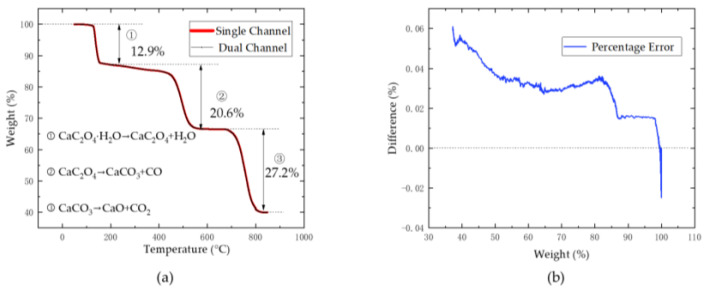
Double-channel differential comparison experimental results. (**a**) Thermogravimetric curves of CaC_2_O_4_·H_2_O using single-channel and dual-channel systems. (**b**) Error between the curves obtained from the single-channel and dual-channel testing methods in (**a**).

**Figure 15 micromachines-14-00823-f015:**
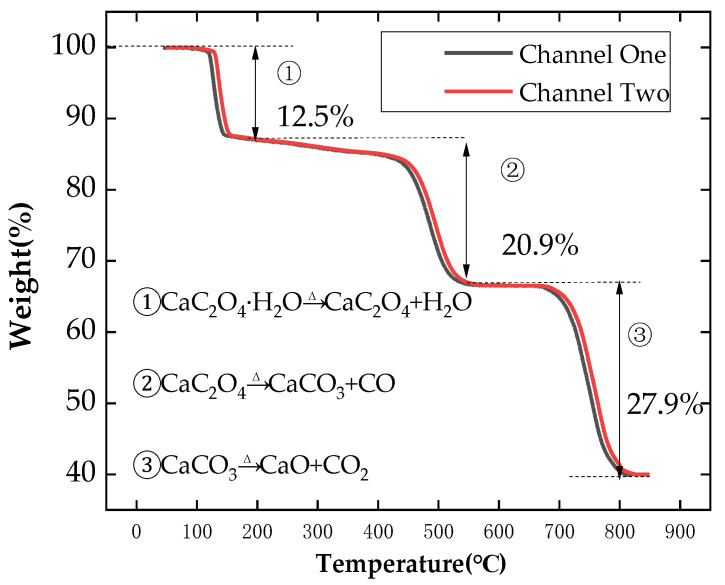
Double-channel thermogravimetric curve of CaC_2_O_4_·H_2_O.

## Data Availability

The data presented in this study are available on request from the corresponding author.

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
