# Peer review of "An Engineering Method for Resonant Microcantilever Using Double-Channel Excitation and Signal Acquisition Based on LabVIEW"

_micromachines, 2023, doi:10.3390/mi14040823_

Round 1
Reviewer 1 Report
This manuscript reports a dual-channel testing system for microcantilevers, which can test two microcantilevers simultaneously to improve detection efficiency. The work introduces the system in detail and verifies it through experiments. The results are interesting and the manuscript was well-organized. Some minor revisions and editing are suggested:
1. ‘Thermalgravimetric’ should be ‘thermogravimetric’ in the abstract section.
2. Some of the words in the keywords section do not need to be capitalized.
3. The format of ‘f’ in ‘Based on the time interval f…’ in section 2.3.2 should be changed to italics. The author should also check if there are similar errors in other parts.
4. ‘Shown in Figure.’ at the end of the first paragraph in section 2.3.4 should be ‘shown in Figure 11.’. The second paragraph has problems with both formatting and font.
5. In section 3, the description of experiments (1) and (2) is in the opposite order to that of the subsequent experiments. The author should determine the order of the two experiments correctly.
6. For the true meaning of simultaneously detecting two microcantilever in completely identical environments as mentioned in the article, is it possible to detect two microcantilevers in different environments? This may be more meaningful for detecting the changes of different materials under different conditions.
Author Response
Dear reviewer, thank you for reviewing our manuscript and providing valuable feedback. We appreciate your suggestions, which are of great significance to our research. We have provided the following point-to-point answers based on the questions you have raised.
Q1: ‘Thermalgravimetric’ should be ‘thermogravimetric’ in the abstract section.
A1: We have changed “Thermalgravimetric” to “thermogravimetric” in the manuscript.
Q2: Some of the words in the keywords section do not need to be capitalized.
A2: We have corrected the capitalization of the keywords that should not have been capitalized.
Q3: The format of ‘f’ in ‘Based on the time interval f…’ in section 2.3.2 should be changed to italics. The author should also check if there are similar errors in other parts.
A3: We have made changes in the manuscript as requested.
Q4: ‘Shown in Figure.’ at the end of the first paragraph in section 2.3.4 should be ‘shown in Figure 11.’. The second paragraph has problems with both formatting and font.
A4: We have made changes in the manuscript as requested.
Q5: In section 3, the description of experiments (1) and (2) is in the opposite order to that of the subsequent experiments. The author should determine the order of the two experiments correctly.
A5: We have reordered the experimental procedures in the manuscript as necessary.
Q6: For the true meaning of simultaneously detecting two microcantilever in completely identical environments as mentioned in the article, is it possible to detect two microcantilevers in different environments? This may be more meaningful for detecting the changes of different materials under different conditions.
A6: The system designed in this study can fulfill the requirement of simultaneously detecting two microcantilever in two different environments. By utilizing two reaction chambers, users can configure different experimental conditions to accomplish real-time monitoring of the two microcantilever in different environments.
Reviewer 2 Report
This work presents a dual channel resonant microcantilever based test system using labVIEW. The effectiveness is clearly demonstrated by two experiments. It is a typical application research. Although the novelty is not high, this work is also encouraged.
Author Response
Dear reviewer, thank you very much for reviewing our manuscript, and we also appreciate your recognition and encouragement of our work.
Reviewer 3 Report
Overview: The authors presented an engineering method to process a thermalgravimetric test on resonant cantilevers by the use of LABVIEW and National Instrument DAQ devices. The method was named as dual-channel testing method. The experimental data showed that the approach is effective and produce good results.
However, there are limitations regarding to the results.
1. There are lacks of published reports and related documents relating to the method based on LABVIEW. The main reason is that the method is just another engineering approach to conduct the experiment. It is totally not a new method at all. Therefore, I suggest to change the title and the writing approach, such as: An engineering method for resonant microcantilever using double-channel excitation based on LabView” or equivalent expression.
2. The same reason as #1, then most of the citations are not related to the core of the paper which is about the engineering approach based on Labview. The papers cited in this manuscript are about the either cantilevers or the contribution of the thermalgravimetric test. The authors are recommended to add more relevant publications or reports related to this topic.
3. The authors had better to compare the results from the new approach with respect to the current method to claim the advances in the aspect of efficiency, time, results …
4. The paper presents about the graphical code based on Labview. It is difficult to review such type of content without the code. So I recommend the authors provide the code the check the validity of the approach.
Author Response
Dear reviewer, thank you for reviewing our manuscript and providing valuable feedback. We greatly appreciate your suggestions, as they are of significant importance to our research.
Q1: There are lacks of published reports and related documents relating to the method based on LABVIEW. The main reason is that the method is just another engineering approach to conduct the experiment. It is totally not a new method at all. Therefore, I suggest to change the title and the writing approach, such as: An engineering method for resonant microcantilever using double-channel excitation based on LabView” or equivalent expression.
A1: We have made changes to the title to “An Engineering Method for Resonant Microcantilever Using Double-Channel Excitation and Signal Acquisition Based on LabVIEW”.
Q2: The same reason as #1, then most of the citations are not related to the core of the paper which is about the engineering approach based on LabVIEW. The papers cited in this manuscript are about the either cantilevers or the contribution of the thermalgravimetric test. The authors are recommended to add more relevant publications or reports related to this topic.
A2: We have added relevant literature and additional explanations pertaining to LabVIEW in the third and fourth paragraphs of the abstract, and in sections 2.2.1, 2.3.1, 2.3.3.
Q3: The authors had better to compare the results from the new approach with respect to the current method to claim the advances in the aspect of efficiency, time, results …
A3: We have included supplementary information in the conclusion section.
Q4: The paper presents about the graphical code based on LabVIEW. It is difficult to review such type of content without the code. So I recommend the authors provide the code the check the validity of the approach.
A4: Because a company is negotiating with us to purchase the presented dual-channel testing system, we are unable to provide the code. We apologize for any inconvenience this may cause. We have added content related to the UI interface of the system in the revised manuscript instead. We sincerely hope for your understanding!
Round 2
Reviewer 3 Report
The revised manuscript is recommended for publication.